# Usefulness of Preoperative Determination of Serum MR-ProAdrenomedullin Levels to Predict the Need for Postoperative Organ Support in Abdominal Oncological Surgery

**DOI:** 10.3390/jpm13071151

**Published:** 2023-07-18

**Authors:** Fernando Ramasco Rueda, Antonio Planas Roca, Rosa Méndez Hernández, Angels Figuerola Tejerina, Eduardo Tamayo Gómez, Carlos Garcia Bernedo, Emilio Maseda Garrido, Natalia F. Pascual Gómez, Olga de la Varga-Martínez

**Affiliations:** 1Department of Anaesthesiology and Surgical Intensive Care, Hospital Universitario de la Princesa, Diego de León 62, 28006 Madrid, Spain; antonioplanas@hotmail.com (A.P.R.); rosamen2004@hotmail.com (R.M.H.); 2Departament of Preventive Medicine and Public Health, Hospital Universitario de la Princesa, Diego de Leon 62, 28006 Madrid, Spain; angels.figuerola@gmail.com; 3Teaching Unit of Anesthesiology and Critical Pathology, Faculty of Medicine, University of Valladolid, 47003 Valladolid, Spain; eduardo.tamayo@uva.es; 4Departament Anesthesiology and Surgical Intensive Care, Valladolid University Clinical Hospital, 47003 Valladolid, Spain; 5Department of Anaesthesiology and Surgical Intensive Care, Hospital del Mar, Passeig Marítim 25-29, 08003 Barcelona, Spain; cagbernedo@hotmail.com; 6Department of Anaesthesiology and Surgical Intensive Care, Hospital QuirónSalud Valle del Henares, Constitution Avenue, 249, Torrejon de Ardoz, 28850 Madrid, Spain; emilio.maseda@gmail.com; 7Departament of Clinical Analysis, Hospital Universitario de la Princesa, Diego de Leon 62, 28006 Madrid, Spain; nataliapascualgomez@gmail.com; 8Department of Anaesthesiology, Infanta Leonor University Hospital, Gran Via del Este 80, 28031 Madrid, Spain; olga.v.m91@gmail.com

**Keywords:** biomarkers, MR-ProADM, prodrenomedullin, organic support, perioperative risk

## Abstract

The need for postoperative organic support is associated with patient outcomes. Biomarkers may be useful for detecting patients at risk. MR-ProADM is a novel biomarker with an interesting profile that can be used in this context. The main objective of this study was to verify whether there was an association between the preoperative serum levels of MR-ProADM and the need for organic support after elective abdominal cancer surgery, and to determine the preoperative MR-ProADM value that predicts the need for postoperative organic support. This was a multicenter prospective observational study conducted by four tertiary hospitals in Spain between 2017 and 2018. Plasma samples were collected for the quantification of MR-ProADM from adults who underwent major abdominal surgery during 2017–2018. The primary outcome was the need for organic support in the first seven postoperative days and its association with the preoperative levels of MR-ProADM, and the secondary outcome was the preoperative levels of MR-ProADM in the study population. This study included 370 patients with a mean age of 67.4 ± 12.9 years. Seventeen percent (63 patients) required some postoperative organic support measures in the first week. The mean preoperative value of MR-ProADM in patients who required organic support was 1.16 ± 1.15 nmol/L. The AUC-ROC of the preoperative MR-ProADM values associated with the need for organic support was 0.67 (95% CI: 0.59–0.75). The preoperative MR-ProADM value, which showed the best compromise in sensitivity and specificity for predicting the need for organic support, was 0.70 nmol/L. The negative predictive value was 91%. A multivariate analysis confirmed that a preoperative level of MR-ProADM ≥ 0.70 nmol/L is an independent factor associated with risk of postoperative organic support (OR 2, 6). Elevated preoperative MR-ProADM levels are associated with the need for postoperative organic support. Therefore, MR-ProADM may be a useful biomarker for perioperative risk assessment.

## 1. Introduction

Based on data collected from patients after non-cardiac surgery, the European Surgical Outcomes Study (EuSOS) reported a postoperative mortality rate of 4% [1]. In most cases, perioperative mortality was attributed to cardiovascular complications and sepsis [2]. Up to 9% of patients undergoing major abdominal surgery experience organ failure in the postoperative period, which is correlated with an increase in length of stay in an intensive care unit (ICU) and higher mortality [3].

The preoperative identification of at-risk patients is crucial for proposing evidence-based selective actions aimed at reducing the incidence of complications and mortality. The most widespread practice for identifying at-risk patients is the use of prognostic indices and preoperative risk-assessment scales [4].

However, there may be a “clinically silent” disease that is reflected only by elevated plasma biomarker levels and is associated with increased mortality. Preoperative determination of these biomarkers can improve the ability to predict complications, allowing for precise and dynamic risk stratification throughout the perioperative period [5,6].

Current research is directed towards the use of biomarkers [7]. The preoperative determination of brain natriuretic peptides (ProBNP) is recommended in the main cardiovascular risk guidelines for noncardiac surgery [8]. Álvarez-Zurro et al. reported that preoperative values of ProBNP greater than 300 pg/mL are associated with a higher probability of cardiovascular complications and mortality after non-cardiac surgery [9].

Regional-ProAdrenomedullin (MR-ProADM) is a pro-peptide biomarker of adrenomedullin, a peptide synthesized in several tissues, including the bone, adrenal cortex, kidney, lung, heart, and vessels. The biological effects include vasodilation, inotropism, natriuresis, and bronchodilation [10].

MR-ProADM is a marker of the severity of a wide spectrum of pathological conditions (sepsis, chronic obstructive pulmonary disease, and renal failure, etc.). Elevated plasma levels are associated with hemodynamic and cardiovascular dysfunctions [11]. Interest in this biomarker lies in its role as a marker of microcirculatory and organ dysfunctions, which can discriminate between patients with lower life expectancies in heterogeneous populations [12,13] and, in turn, predict and stratify organ failure [14].

The usefulness of MR-ProADM as a predictor of morbidity and mortality in surgical patients has yet to be elucidated.

The main objective of this study was to verify whether there is an association between the preoperative serum levels of MR-ProADM and the need for postoperative organic support (POS) after intermediate- and high-risk elective abdominal cancer surgery, and to establish a preoperative MR-ProADM value with the best sensitivity and specificity compromise to predict the need for POS.

The secondary objectives were to characterize the preoperative levels of MR-ProADM in a population of patients scheduled for intermediate- and high-risk abdominal cancer surgery, and to determine the association between MR-ProADM and perioperative variables such as age, ASA classification, Revised Cardiac Risk Index (RCRI), POSSUM, and Surgical Apgar Score (SAS) index.

## 2. Materials and Methods

### 2.1. Study Population

This multicenter, prospective, observational study was conducted between February 2017 and May 2018 at the University Hospital of La Princesa (Madrid, Spain), Hospital Clínico Universitario (Valladolid), Hospital del Mar (Barcelona), and Hospital Universitario de la Paz (Madrid).

This study recruited consecutive adult patients older than 18 years scheduled for high- and intermediate-risk elective oncological abdominal surgery, according to the Guidelines on Cardiovascular Surgery [15].

Exclusion criteria were patients undergoing low-risk abdominal surgery and patients who did not provide informed consent.

A pilot study was conducted at the University Hospital of La Princesa to assess the capacity of MR-ProADM to predict the risk of needing POS.

Based on the results, it was determined that it was necessary to recruit a minimum of 55 patients in each group, accepting an α risk of 0.05, and a capacity of 80% to detect differences.

### 2.2. Measurement of MR-ProADM

Pre-operative plasma samples were collected before anesthesia induction to quantify MR-ProADM. Measurement was made with TRACE technology (Time Resolved Amplified Cryptate Emission) using a new immunoassay sandwich (Kryptor Compact Plus Analyzer, BRAHMS, Hennigsdorf Germany), with a detection limit of 0.05 nmol/L [16].

All samples were analyzed in the Clinical Analysis Laboratory of the University Hospital of La Princesa. The samples obtained at the Hospital del Mar, Clinical Hospital, University of Valladolid, and Hospital Universitario La Paz were frozen and subsequently transferred to the University Hospital of La Princesa to determine the levels of MR-ProADM.

### 2.3. Study Variables

We collected preoperative clinical and epidemiological variables including age, sex, ASA classification, RCRI [17], POSSUM Score [18], and SAS score [19]. Postoperative variables included the appearance of postoperative organ failure in the first seven postoperative days, hospital stay, and mortality at 30 days and 6 months.

Postoperative organ failure was defined as the presence of at least one of the following conditions: need for administration of amines after volume resuscitation for at least 12 h, need for invasive or non-invasive ventilation not related to sedation and for at least 12 h, and need for new requirements of RRT (renal replacement therapy). The length of the hospital stay and mortality during the first 30 postoperative days were recorded.

### 2.4. Statistic Analysis

For the descriptive analysis of qualitative variables, frequencies were calculated, and Pearson’s χ^2^ test or Fisher’s exact nonparametric test were used for comparisons. For quantitative variables, the mean was calculated with its standard deviation (SD), and comparisons were made using Student’s t-test or the nonparametric Mann–Whitney U test.

To select the best discriminatory value of MR-ProADM for the diagnosis of postoperative organ failure, the areas under the ROC curves (AUC) of the mean, median, and mode were compared. The sensitivity, specificity, positive predictive value (PPV), negative predictive value (NPV), positive likelihood ratio (PVR), negative likelihood ratio (NRV), and probability of correct diagnosis were calculated for MR-ProADM with a higher AUC.

Finally, using the selected MR-ProADM values, an explanatory model of postoperative organ failure was constructed using a multivariate logistic regression analysis. Based on a saturated model, a backward phasing strategy was used until the final model was reached, and odds ratios (OR) corresponding to the model variables were calculated. Goodness of fit was assessed using the Hosmer–Lemeshow test.

Data analysis was carried out using the statistical software SPSS version 19.0 for Windows and EPIDAT version 4.2. Differences were considered statistically significant at *p* < 0.05.

## 3. Results

### 3.1. Descriptive Data

A total of 373 patients were included in this study. Three patients were excluded because a sample analysis was impossible, leaving 370 valid patients who met the criteria and from whom complete data were collected.

The mean age of the patients was 67.4 ± 12.9 years, and 62.2% (230) were older than 65 years. The sample was 65.4% (242) male and 34.6% (128) female. The mean duration and frequency depending on the type of surgical procedure performed are shown in (Table 1).

Regarding the risk scales analyzed, 57.3% (212) presented a score of ≤ 2 in the preoperative ASA classification, whereas in 42.7% (158), the score was ≥ 3. The preoperative RCRI score was 1 point in 77.6% of the patients (287) and ≥ 2 points in the remaining 22.8% (83). The mean POSSUM score was 29,74 points. In total, 49.5% of patients (183) had a score higher than 28 points on this scale. Regarding the SAS score, 8.6% (32) had a score ≤ 4, while 81.4% (338) had a score ≥ 5 points.

The mean length of hospital stay was 16.5 ± 33 days. The mortality rate during the first 30 postoperative days was 8.6% (N = 32). Seventeen percent (n = 63) required organ support measures in the first week after surgery. The incidence in the total population, according to the type of POS measure used, is shown in (Table 2).

### 3.2. Outcome Data (Primary Outcome)

The mean preoperative value of MR-ProADM in patients who required POS was 1.16 ± 1.15 (nmol/L), which was significantly higher than that of those who did not require it, with a mean value of 0.74 ± 0.46 (nmol/L) (*p* < 0.05).

The area under the ROC curve (AUC-ROC) of the preoperative MR-ProADM values associated with the need for POS was analyzed, obtaining an AUC-ROC of 0.67 (95% CI: 0.59–0.75) (Figure 1).

The preoperative MR-ProADM value, which showed the best compromise in sensitivity and specificity in the AUC-ROC for predict the need for POS, was 0.70 nmol/L.

The analysis of the diagnostic characteristics of preoperative levels of MR-ProADM ≥ 0.70 nmol/L is described in detail in Table 3.

With the diagnostic characteristics described in Table 3, the “probability of success” to predict the need for POS in patients with a preoperative MR-ProADM value ≥ 0.70 nmol/L was 62% (95% CI: 56.43–66.56%).

For the MR-ProADM value ≥ 0.70 nmol/L as the cut-off point, the negative predictive value obtained was 91%. None of the patients with preoperative MR-ProADM value ≤ 0.25 nmol/L required POS.

In the univariate analysis performed to determine perioperative variables associated with need for POS, preoperative MR-ProADM levels ≥ 0.70 nmol/L, age, type of surgery, POSSUM Index, and SAS score were significantly associated with POS (Table 4).

To adjust for possible confounding factors, a multivariate analysis was performed using logistic regression. This analysis included all perioperative variables that showed a statistically significant association with the need for POS in univariate analysis. The results are presented in Table 5.

The multivariate analysis confirmed that the preoperative level of MR-ProADM ≥ 0.70 nmol/L was found to be an independent factor associated with the risk of needing POS; these patients had a 2.6-times-higher risk of POS than those with lower levels. This association was independent of age, sex, type of surgery, and the perioperative risk scale score.

### 3.3. Outcome Data (Secondary Outcome)

The mean preoperative value of MR-ProADM in the 370 patients studied was 0.81 ± 0.65 nmol/L, with a median of 0.66 nmol/L and an interquartile range (IQR) of 0.33. In total, 7.6% (28) of patients had a preoperative MR-ProADM value < 0.4 nmol/L.

The relationship of the cut-off points of MR-ProADM ≥ 0.70 nmol/L with the perioperative variables was analyzed. The mean age of patients with MR-ProADM levels ≥ 0.70 nmol/L was higher than that of patients with MR-ProADM < 0.70 nmol/L (*p* < 0.05). Likewise, a significant association (*p* < 0.05) was found between preoperative MR-ProADM levels ≥ 0.70 nmol/L and the following risk scales: RCRI ≥ 2, ASA ≥ 3, and Possum index ≥ 28. Table 6.

To adjust for possible confounding factors, and thus determine if there was an association between MR-ProADM values ≥ 0.70 nmol/L and perioperative variables, a multivariate analysis was performed using logistic regression. All variables found to have a statistically significant association in the univariate analysis were included in the analysis. Table 7.

The logistic regression analysis found that the presence of preoperative MR-ProADM levels ≥ 0.70 nmol/L was independently associated with age ≥ 65 years (OR 3; CI 1.85–4.99; *p* < 0.05), with ASA value ≥ 3 (OR 1.9; CI 1.11–0.09; *p* < 0.05), and with Possum Index ≥ 28 (OR 2.0; CI 1.26–3.30; *p* < 0.05).

## 4. Discussion

Our results confirm the association between preoperative serum levels of MR-ProADM and the need for POS in the first seven days after intermediate- and high-risk abdominal oncological surgery. The preoperative value of MR-ProADM with the best compromise between sensitivity and specificity when predicting the need for POS was 0.7 nmol/L. This is the first prospective multicenter study to establish the prognostic value of MR-ProADM for predicting the need for POS.

With an average age of 67 years, which is higher than that described in European perioperative studies [1], our sample reflects the age associated with oncological surgery and the demographic characteristics of our area of influence. On the other hand, in the clinical scales used for preoperative evaluation in the patients in our series, it was noteworthy that 43% of them presented an ASA classification score ≥ 3, compared to the surgical population of the EUSOS study, in which the percentage of patients with an ASA score ≥ 3 was only 28.8%8 [1]. Likewise, a preoperative RCRI score of ≥ 2 points in 22.8% of the patients in our series contributed to the description of perioperative risk in the oncological surgical population analyzed. These characteristics indicate an increased probability of postoperative complications in the study population.

An interesting aspect of our study was the characterization of the preoperative values of MR-ProADM in an oncological surgical population. The MR-ProADM values considered normal in the healthy population were established as being within a range of 0.10 and 0.64 nmol/L, with a mean value of 0.33 nmol/L [20]. These values were seen to increase with age, establishing as normal a mean value of 0.41 nmol/L for over 55 years. In high-risk patients for cardiac surgery, a preop average value of 0.62 nmol/L was described [21].

In our sample, the mean preoperative value of MR-ProADM was 0.81 nmol/L and, interestingly, only 7.6% of patients in the series presented values below 0.41 nmol/L. Elevated levels of MR-ProADM were associated with cardiovascular disease [22]. and oncological diseases [23]. In a study by Al Shuaibi et al. [24] the mean value of MR-ProADM in oncohematologic patients at a risk of infection was 0.68 nmol/L. In a study by Pavo et al. [23] on the usefulness of different biomarkers to predict cardiovascular complications in patients affected by different types of cancer, the mean value of MR-ProADM ranged between 0.49 and 0.62 nmol/L, increasing as the stage of the oncological disease became more advanced. The association between oncological disease and elevated MR-ProADM levels observed in our study may be due to the fact that the oncological disease itself causes a silent alteration of the cardiovascular system, due to the endothelial dysfunction and inflammation associated with oncological disease.

Patients undergoing major surgery experience surgical trauma and an inflammatory response, which can be associated with the patient’s baseline conditions and comorbidities, and the failure of one or more organs requiring temporary support [25].

In the present study, the incidence of POS was 17%. Dale et al. [3] found that, after colorectal surgery, 9% of patients received POS, and that this need was significantly correlated with a higher incidence of postoperative complications. The difference in the incidence of POS in our series may be mainly due to differences in the types of surgery studied, as our sample included 35% of non-colorectal high-risk surgeries, since it was in these types of high-risk surgeries that we found a higher incidence of POS (32%) compared to colorectal surgery (9%). The importance of anticipating the need for POS lies in the management of material and human resources, and the prolongation of a hospital stay and its association with mortality.

In our study, the mean preoperative serum levels of MR-ProADM in patients who required POS was 1.16 nmol/L, which was significantly different to the mean value of 0.74 nmol/L in patients without POS. The prognostic value of MR-ProADM has primarily been studied in the context of sepsis. The results of a study by Cain et al. [26] were ratified by other groups, such as Valenzuela et al. [27] and Suberviola et al. [28] and the MR-ProADM values were close to those found to be associated with POS in the present study. The findings in our patients with MR-ProADM levels similar to those of other patients suggest the existence of silent inflammation and endothelial alterations in the perioperative period, which are responsible for organ failure, and the consequent need for POS [29].

To assess the discriminative capacity as a diagnostic test to predict the need for POS, we estimated the AUC-ROC of the preoperative MR-ProADM in our sample, which was of moderate predictive accuracy with a value of 0.67 (95% CI: 0.59–0.75).

Several studies of scales and biomarkers accepted as risk predictors in other scenarios showed AUC-ROC values similar to our sample [30].

A recent study by Andrés et al. [31] analyzed the ability of MR-ProADM and other markers to detect organ failure in patients with severe infections. In that study, the AUC-ROC of the MR-ProADM for the diagnosis of organ failure was 0.79 (0.72–0.86), which is considered a very good predictive capacity and is above that obtained with other biomarkers studied in the same work.

In our sample, the preoperative MR-ProADM value with the best sensitivity and specificity compromise to predict the need for POS was 0.70 nmol/L. In the multivariate analysis, the OR for the prediction of POS with a preoperative MR-ProADM value equal to or greater than 0.70 nmol/L was 2.6. It is important to highlight that the MR-ProADM value < 0.70 nmol/L provided a negative predictive value greater than 90%, which allows us to affirm that, in more than 90% of cases, patients with a preoperative MR-ProADM value < 0.70 nmol/L will not require POS. No patient in the sample with a preoperative MR-ProADM value ≤ 0.25 nmol/L required POS in our series, indicating that healthy patients have a very low probability of complications.

Bernal-Morell et al. evaluated the determination of a predictive cut-off point for MR-ProADM in the prediction of organ failure and mortality in patients with sepsis [32]. The cutoff point of MR-ProADM to detect organ failure in this context was 1.8 nmol/L, with a negative predictive value of 54%. In the study by Andaluz et al. [33], also in septic patients, the presence of an MR-ProADM value of 1.79 nmol/L predicted higher mortality. An MR-ProADM value less than 0.88 nmol/L ruled out mortality in the 28 days after admission to the ICU. Also noteworthy in this regard is a study by Vigué et al. on the predictive capacity of MR-ProADM, which predicted a SOFA > score of 9 with a cut-off point of 1.03 nmol/L [34].

In published studies referring to the predictive value of the perioperative values of MR-ProADM, the following cut-off points were obtained:○A preoperative value of 0.87 nmol/L in the pilot study of our group to predict the need for POS [35];○A preoperative value of 0.85 nmol/L in the study by Golubović et al. to predict postoperative mortality in abdominal surgery [36];○A preoperative value of 0.77 ± 0.37 nmol/L in the study of Egyed et al. [37];○A preoperative value of 1.3 nmol/L in the study referring to the prognostic capacity of mortality after TAVI [38];○A postoperative value of 3.2 nmol/L in the study for the prediction of mortality in the postoperative period of cardiac surgery [39].

In the study population, the presence of MR-ProADM levels ≥ 0.70 nmol/L was independently associated with age > 65 years, ASA values ≥ 3, and a POSSUM score ≥ 28. This association is consistent with the idea that the worse the patient’s clinical status, the more likely it is that the biomarker is elevated, and that it is possibly related to a basal inflammatory state and endothelial dysfunction [40].

The results of this study highlight the usefulness of MR-ProADM as a global risk biomarker in the perioperative period after major abdominal oncological surgery.

The profile previously demonstrated by MR-ProADM in scenarios and pathologies other than the surgical patient, shows its usefulness in the prediction and stratification of perioperative risk, resulting in a marker that predicts the absence of need for POS in patients with preoperative values < 0.70 nmol/L, presenting a high negative predictive value (NPV) [41,42,43,44].

### Study Limitations

This study has several limitations regarding the usefulness of MR-ProADM for the prediction of POS in the postoperative period after major abdominal oncological surgery. This was an observational study with inherent limitations. Although this was a multicenter study, there were differences in patient recruitment, with almost half originating from a single center, a minority from another, and almost the other half distributed among the other two participating centers.

These limitations, in addition to the possibilities offered by this study, imply the need for more studies to determine the role of MR-ProADM in perioperative risk stratification.

## 5. Conclusions

Elevated preoperative MR-ProADM levels are associated with the need for POS. The preoperative value of the MR-ProADM, with the best compromise in sensitivity and specificity for predicting the need for POS, was 0.7 nmol/L. Preoperative values of MR-ProADM lower than 0.70 nmol/L significantly rule out the need for POS. Preoperative levels of MR-ProADM ≥ 0.70 nmol/L were independently associated with other variables related to increased perioperative risk.

## Figures and Tables

**Figure 1 jpm-13-01151-f001:**
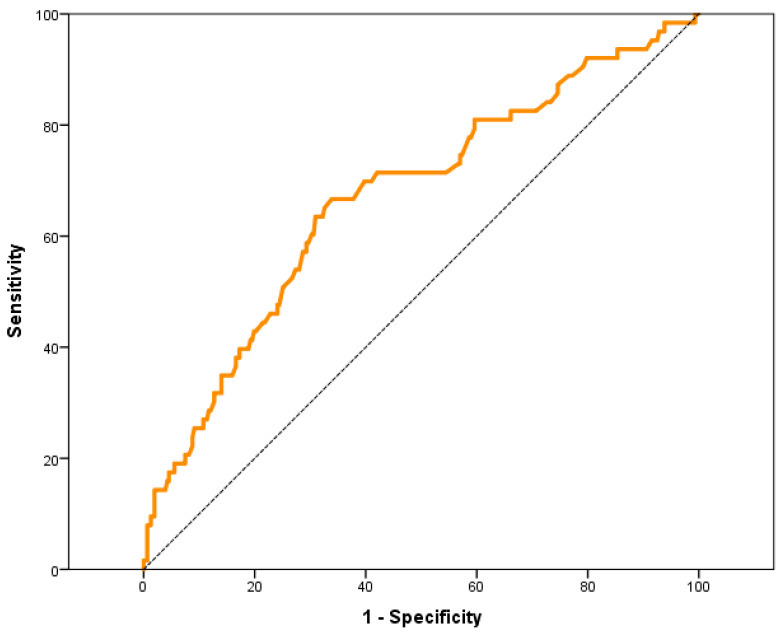
ROC curve of preoperative values of MR-ProADM with respect to the need for PCOS.

**Table 1 jpm-13-01151-t001:** Surgical procedures: frequency and average duration of intervention.

Surgery Group	Frequency (n)	Average Duration (Minutes)
Colorectal	227	211.4 ± 78
Pancreatohepatobiliary	69	313.0 ± 139
Esophagogastroduodenal	29	261.7 ± 134
Cystectomy	30	267.4 ± 80
Open Nephrectomy	15	154.1 ± 44

**Table 2 jpm-13-01151-t002:** Frequency and percentage of the need for POS.

Postoperative Organic Support	Frequency (n)	Percentage (%)
Vasoconstrictors only	31	8.4%
Mechanical ventilation only	10	2.7%
Renal replacement therapy only	0	0%
Vasoconstrictors + Mechanical ventilation	16	4.3%
Vasoconstrictors + Renal replacement therapy	1	0.3%
Mechanical ventilation + Renal replacement therapy	1	0.3%
Vasoconstrictors + Mechanical ventilation + Renal replacement therapy	4	1.1%

**Table 3 jpm-13-01151-t003:** Diagnostic characteristics of MR-ProADM ≥ 0.70 nmol/L.

Diagnostic Characteristics of MR-ProADM ≥ 0.70 nmol/L	Statistical Value	IC 95%	
		Lower Limit	Upper Limit
Sensitivity	69.84%	56.82%	80.43%
Specificity	59.93%	54.20%	65.42%
Positive Predictive Value	26.35%	19.98%	33.82%
Negative Predictive Value	90.64%	85.56%	94.12%
Positive Likelihood Ratio	1.74	1.41	2.16
Negative Likelihood Ratio	0.50	0.34	0.74
Probability of Correct Diagnosis	62%	56.4%	66.56%

**Table 4 jpm-13-01151-t004:** Univariate analysis of the need for POS with respect to the perioperative variables studied (demographics, surgeries, scales, and MR-ProADM).

	OR	IC 95%	*p*
Age ≥ 65 years	1.7	1.00–2.76	<0.05
Sex	0.6	0.38–1.09	ns
Type of surgery	5.5	2.20–5.65	<0.05
ASA ≥ 3	1.5	0.94–2.31	ns
RCRI ≥ 2	1.5	0.92–2.41	ns
POSSUM ≥ 28	4.3	2.40–7.87	<0.05
SAS ≤ 4	3.9	2.56–5.95	<0.05
MR-ProADM ≥ 0.70 nmol/L	2.8	1.71–4.63	<0.05

ASA: American Society of Anesthesiologists; RCRI: Revised Cardiac Risk Index, POSSUM: Physiological and Operative Severity Score for the Enumeration of Mortality and Morbidity; SAS: Surgical Apagar Score. ns: no statistical significance.

**Table 5 jpm-13-01151-t005:** Multivariate analysis of the need for POS with respect to the perioperative variables studied (demographics, surgeries, scales, and MR-ProADM).

	OR	IC 95%	*p*
Age ≥ 65 years	1.0	0.49–2.07	ns
Sex	0.7	0.35–1.40	ns
Type of surgery	1.2	0.98–1.58	ns
ASA ≥ 3	0.8	0.41–1.65	ns
RCRI ≥ 2	0.9	0.42–1.84	ns
POSSUM ≥ 28	4.4	2.09–9.16	<0.05
SAS ≤ 4	5.3	2.30–12.4	<0.05
MR-ProADM ≥ 0.70 nmol/L	2.6	1.33–5.09	<0.05

ASA: American Society of Anesthesiologists; RCRI: Revised Cardiac Risk Index, POSSUM: Physiological and Operative Severity Score for the Enumeration of Mortality and Morbidity; SAS: Surgical Apagar Score. ns: no statistical significance.

**Table 6 jpm-13-01151-t006:** Univariate Analysis (expressed in OR). MR-ProADM ≥ 0.70 nmol/L with respect to the variables studied.

	OR	IC 95%	*p*
Age ≥ 65 years	1.7	1.42–1.96	<0.05
Sex	0.9	0.71–1.15	ns
Type of surgery	1.0	0.79–1.27	ns
ASA ≥ 3	1.9	1.48–2.34	<0.05
RCRI ≥ 2	1.8	1.46–2.20	<0.05
POSSUM ≥ 28	1.9	1.50–2.45	<0.05
SAS ≤ 4	1.3	0.92–1.77	ns

ASA: American Society of Anesthesiologists; RCRI: Revised Cardiac Risk Index, POSSUM: Physiological and Operative Severity Score for the Enumeration of Mortality and Morbidity; SAS: Surgical Apagar Score. ns: no statistical significance.

**Table 7 jpm-13-01151-t007:** Multivariate analysis of preoperative levels of MR-ProADM ≥ 0.70 nmol/L with respect to the variables studied (demographics, surgeries, and scales).

	OR	IC 95%	*p*
Age ≥ 65 years	3.0	1.85–4.99	<0.05
Sex	1.2	0.71–1.93	ns
Type of surgery	1.2	0.95–1.42	ns
ASA ≥ 3	1.9	1.11–3.09	<0.05
RCRI ≥ 2	1.6	0.88–2.98	ns
POSSUM ≥ 28	2.0	1.26–3.30	<0.05
SAS ≤ 4	1.3	0.59–2.96	ns

ASA: American Society of Anesthesiologists; RCRI: Revised Cardiac Risk Index, POSSUM: Physiological and Operative Severity Score for the Enumeration of Mortality and Morbidity; SAS: Surgical Apagar Score. ns: no statistical significance.

## Data Availability

The data presented in this study are available on request from the corresponding author. The data are not publicly available due to privacy.

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
