# Peer review of "Usefulness of Preoperative Determination of Serum MR-ProAdrenomedullin Levels to Predict the Need for Postoperative Organ Support in Abdominal Oncological Surgery"

_jpm, 2023, doi:10.3390/jpm13071151_

Round 1
Reviewer 1 Report
I’ve had the opportunity to revise the manuscript entitled “USEFULNESS of preoperative determination of serum MR-2 ProAdrenomedullin levels to predict the need for postoperative 3 organ support in ABDOMINAL oncological SURGERY.”. In this multicenter prospective study, the authors recruited 370 oncological surgical patients and measured preoperative levels of MR-ProADM. Then, they identified the predictive capacity and relevant cut-off values of MR-ProADM on postoperative organ support.
I must commend the authors for performing such a robust study, with a high number of patients, multicenter in nature, and thorough statistical analysis, with both univariate and multivariate assessments. The manuscript is easy to read and is coherent regarding its objectives. The discussion also provides context and compares with existing literature from other contexts where MR-ProADM has been used as a prognostic factor. Both the discussion and the conclusion are balanced.
I have the following minor commentaries that could help enrich the manuscript and improve the reading flow:
Introduction, line 70 “The mortality rate associated with this surgery remains unknown.”.
- Since this is the first line from the manuscript, it should mention which surgery it refers to (i.e., abdominal oncological surgery).
Materials and Methods, line 118-119: “This study recruited consecutive adult patients older than 18 years scheduled for 118 high- and intermediate-risk abdominal surgery”.
- Authors should provide further detail that it is oncological surgery as an inclusion criteria.
- Also, they should state if it included only elective surgery or also elective + emergency surgeries.
Methods
- Line 146: It seems that Renal Replacement therapy (as an outcome) refers to new requirements of RRT. It should be clarified if it considers chronic requirements + new or only new requirements of RRT.
Discussion
- Line 341-343 Bernal-Morell et al. evaluated the determination of a predictive cut-off point for MR-341 ProADM to predict organ failure and mortality in patients with sepsis by Bernal-Morell 342 et al.33
o The name of the 1st authors of the paper is duplicated, please revise.
no
Author Response
I’ve had the opportunity to revise the manuscript entitled “USEFULNESS of preoperative determination of serum MR-2 ProAdrenomedullin levels to predict the need for postoperative 3 organ support in ABDOMINAL oncological SURGERY.”. In this multicenter prospective study, the authors recruited 370 oncological surgical patients and measured preoperative levels of MR-ProADM. Then, they identified the predictive capacity and relevant cut-off values of MR-ProADM on postoperative organ support.
I must commend the authors for performing such a robust study, with a high number of patients, multicenter in nature, and thorough statistical analysis, with both univariate and multivariate assessments. The manuscript is easy to read and is coherent regarding its objectives. The discussion also provides context and compares with existing literature from other contexts where MR-ProADM has been used as a prognostic factor. Both the discussion and the conclusion are balanced.
Hello. Many thanks you so much for your words. The study is the result of a very important work of many people during several years between the design, the preparation and the execution and analysis of the data until reaching here . They are hundreds of hours of work. Thank you so much.
I have the following minor commentaries that could help enrich the manuscript and improve the reading flow:
Introduction, line 70 “The mortality rate associated with this surgery remains unknown.”.
- Since this is the first line from the manuscript, it should mention which surgery it refers to (i.e., abdominal oncological surgery).
We have removed that sentence, and start directly with the quote from the EUSOS study . Perhaps that start could lead to confusion even adding. Thanks for the observation.
Materials and Methods, line 118-119: “This study recruited consecutive adult patients older than 18 years scheduled for 118 high- and intermediate-risk abdominal surgery”.
- Authors should provide further detail that it is oncological surgery as an inclusion criteria.
- Also, they should state if it included only elective surgery or also elective + emergency surgeries.
We have corrected it by adding " elective oncological."
Methods
- Line 146: It seems that Renal Replacement therapy (as an outcome) refers to new requirements of RRT. It should be clarified if it considers chronic requirements + new or only new requirements
Thanks for the observation, we have clarified in the text that they are new requirements of RRT.
Discussion
- Line 341-343 Bernal-Morell et al. evaluated the determination of a predictive cut-off point for MR-341 ProADM to predict organ failure and mortality in patients with sepsis by Bernal-Morell 342 et al.33
The name of the 1st authors of the paper is duplicated, please revise.
We have removed the repetition of the name
Reviewer 2 Report
1) the paper is interesting but the text must be improved as in many parts it is redundant. As an example, I should not use 4 different scoring systems.
2) the reasons for the need of POS shoud be explained (infections? perioperative arterial hypotension or shock? etc)
3) the duration of POS should be explained: as an example, a short term (< 24 hours) vasopressor need is quite different from a week-long one
4) the high preoperative values of MR-Pro ADM in patients requiring POS should be explained at leastst on a hypothetical basis
the quality of English language must be improved
Author Response
- the paper is interesting, but the text must be improved as in many parts it is redundant. As an example, I should not use 4 different scoring systems.
Thank you for evaluating the study as interesting. The study is the result of a very important work of many people during several years between the design, the preparation and the execution and analysis of the data until reaching here . They are hundrends of hours of work. Thank you so much.
In a study like this, a lot of data was collected and precisely for the publication we have strictly chosen those that we have considered necessary and useful for the reader.
The main objective is the evaluation of a Biomarker, MR-ProADM as a predictor of the need for postoperative organic support, and that is the main essence of this multicenter study.
As a possibility of pre-perioperative evaluation variable, it seemed interesting to us to measure the four scales that we think are considered the most useful and used in practice and see their association and also use them in the regression analysis. The 4 scales are the most used and considered in the literature and clinical practice.
We believe that the article is not redundant, and a great effort has been made so that it is not . The other reviewer precisely appreciates the precision and conciseness of the article ( “The manuscript is easy to read and is coherent regarding its objectives. The discussion also provides context and compares with existing literature from other contexts where MR-ProADM has been used as a prognostic factor. Both the discussion and the conclusion are balanced.” ) , which is why we think that as it stands , the review is fine and understandable in our opinion.
Reviewing his review, it seems that it is the introduction that suggests improvement. The introduction begins by briefly planting the serious problem of mortality associated with surgery and the importance of detecting patients at risk,continues to assess the growing importance of biomarkers in clinical practice guidelines, we describe the biomarker under study, and finally plant the objectives of the study.
We do not see redundancy.
Regarding Material and Methods, we respond in points 2 and 3 of your review.
Regarding results and conclusions, it seems that they are ok in their assessment.
We think that although the English could be somewhat improved and we are going to do so, the writing is not redundant, and a great effort has been made to avoid it , and that is how it is valued .
- the reasons for the need of POS shoud be explained (infections? perioperative arterial hypotension or shock? etc)
Thank you very much. It is important to highlight the main objective of the study, which is to investigate whether there is a relationship between the MR-ProADM biomarker and the need for postoperative organic support. As it is well known. and this is how it is described in articles from the citations of our article, the need for postoperative organic support can only be related to the aggression of the surgery itself, and this need for support is relevant not only in terms of mortality, but also also stays and others.
We hope that the description of everything related to postoperative organic support in this sample of patients will be another article as a substudy of this one. We know the complications that in summary were: sepsis (15,1%), Acute Respiratory Distress Syndrome (2,4 %), Acute Renal Failure (11,1%), anastomotic dehiscence (7,8%), Pneumonia 2,2%) , among others.But we think that we should not put them in this article because it would blur the objective of it. If they are ok for the substudy.
Thank you very much.
- the duration of POS should be explained: as an example, a short term (< 24 hours) vasopressor need is quite different from a week-long one
Many Thanks. Related to this topic and due to this suggestion of yours, we better complete the definition that had a need for postoperative organ support in our study:
“Postoperative organ failure was defined as the presence of at least one of the following conditions: need for administration of amines after volume resuscitation, and for at least 12 hours, need for invasive or non-invasive ventilation not related to sedation and at least 12 hours, and need for new requirements of RRT.renal replacement therapy. The length of the hospital stays and mortality during the first 30 postoperative days were recorded.”
It is clear that having the need for support less than 24 hours is not the same as 7 days. And in the aforementioned substudy we will study that relationship as well. But in this case the objective is the relationship of the marker levels with the need for POS defined as such, a definition that meets the standards requested for such and even more in all related studies.
For example, the need for POS in the population of our study has a 7.2 times greater probability of mortality than those who do not require it. It is a very interesting piece of information, and one that we will try to publish in the future, but in this article it blurs the objective already referred to several times in this reply to the reviewer .
4) the high preoperative values of MR-Pro ADM in patients requiring POS should be explained at leastst on a hypothetical basis
All the study, the main hypothesis, and the objectives, and most of the explanation in the discussion is oriented to explain why the elevated levels of MR-ProADM can be associated to POS , and why they can have it elevated .
Specifically, for example, this discussion paragraph explicitly answers this reviewer's question:
“In our sample, the mean preoperative value of MR-ProADM was 0.81 nmol/L and, interestingly, only 7.6% of patients in the series presented values below 0.41 nmol/L. Elevated levels of MR-ProADM have been associated with cardiovascular disease22 and oncological diseases.23 Therefore, in a study by Al Shuaibi et al.,24 the mean value of MR-ProADM in oncohematologic patients at a risk of infection was 0.68 nmol/ L. In a study by Pavo et al25 on the usefulness of different biomarkers to predict cardiovascular complications in patients affected by different types of cancer, the mean value of MR-ProADM ranged between 0.49 and 0.62 nmol/L, increasing as the stage of the oncological disease was more advanced. The association between oncological disease and elevated MR-ProADM levels observed in our study may be due to the fact that the oncological disease itself causes a silent alteration of the cardiovascular system, due to endothelial dysfunction, and inflammation associated with oncological disease.”
We think that this question is well explained in the discussion.
Comments on the Quality of English Language
the quality of English language must be improved
We have done an in-depth review of the English of the paper with MDPI english edition
Round 2
Reviewer 1 Report
the authors have answered my previous minor comments.
I have no further suggestions.
Reviewer 2 Report
the requested adjustements have been performed. No other remarks